| Open Peer Review | Bacteriology | Minireview

# The role of plant host genetics in shaping the composition and functionality of rhizosphere microbiomes

María Negre Rodríguez,[1] Adele Pioppi,[1,2] Ákos T. Kovács[1,2]

**ABSTRACT** The plant microbiome comprises a wide range of microorganisms associated with different plant tissues. Over the past decades, significant research efforts have focused on understanding and harnessing the plant rhizosphere microbiome for optimal plant growth and health. However, while environmental factors are often the primary focus of these studies, the influence of plant genotype remains comparatively underexplored. In fact, the plant genotype influences a multitude of factors including root morphology and exudate composition, which play an integral role in shaping the rhizosphere niche. In this review, we discuss how alterations in plant host genetics could lead to differences in the assembly and diversity of the rhizosphere microbiome. Furthermore, we summarize current approaches to decipher complex plant traits and their ecological implications in host-microbiome systems and interactions.

**KEYWORDS** rhizosphere microbiome, plant genotype, domestication, root architecture, root exudates

As human civilization advanced, agricultural systems evolved alongside it to meet the increasing demands of the growing populations. This led to the selection of crops with desirable traits to increase production. With environmental challenges like drought, pests, and nutrient deficiencies intensifying, the demand for effective crop management strategies has grown accordingly (1). However, the widespread use of pesticides and intensive management practices have negative impacts on the environment, which prompt the search for more sustainable solutions.

Soil hosts a wide and diverse array of microorganisms, fueling growing curiosity about their interactions with plants. The plant microbiome, comprising microbial communities associated with various plant tissues, provides numerous benefits, including pathogen protection, growth promotion, enhanced resilience to drought and nutrient stress, and increased crop yields (2). Given the demonstrated advantages of plant-associated microbes, a new question arises: which factors define a plant microbiome? Detailed dissection of these interactions and understanding how plants regulate the assembly and diversity of their microbiomes have become a key area of research.

A primary focus for manipulating the plant microbiome lies within the rhizosphere, a region of soil surrounding plant roots, which supports a rich diversity of microorganisms (3). It is hypothesized that plants exert genetically encoded control over the assembly and diversity of the rhizosphere microbiome, with factors such as root exudates playing a critical role (3). By coordinating the microbiome, the plant is thought to extend its genome; therefore, the microbiome is depicted as a "second genome" due to its influence on plant growth (2). Different plant genotypes harbor distinct compositions of rhizosphere microbial communities, suggesting an intriguing link between plant genetics and microbiome diversity, which opens the door to promising strategies for enhancing crop performance through microbiome manipulation. This review summari-

**Peer Reviewer** Xiaoping Li, The Pennsylvania State University, State College, Pennsylvania, USA

Address correspondence to Ákos T. Kovács, a.t.kovacs@biology.leidenuniv.nl.

María Negre Rodríguez and Adele Pioppi contributed equally to this article. The author order was determined based on alphabetical order of family names.

The authors declare no conflict of interest.

See the funding table on p. 10.

*[This article was published on 11 July 2025 with a missing reference. The References were corrected in the current version, posted on 14 July 2025.]*

zes various examples of how plant genetics influence its microbiome, with the aim to exploit this fundamental understanding toward sustainable agriculture.

## DOMESTICATION: HOW THE PLANT GENOTYPE INFLUENCES THE RHIZO-SPHERE MICROBIOME

Historically, agricultural efforts have prioritized optimizing crop traits and maximizing yield to meet the demands of growing populations. However, recent research has expanded this focus to the complex interactions between crops and their associated microbiomes. Studies reveal that distinct crop varieties can engage unique microbial communities, underscoring the role of plant genetics in shaping the rhizosphere microbiome. This research opens pathways for targeted agricultural practices, where microbiome management becomes a core component to improve crop health and productivity.

A prominent example of how a plant's evolutionary history shapes its root microbiome is the process of domestication. Most of the plants we commonly use and consume nowadays are the product of artificial selection of desirable traits during the last millennia. Through domestication, the diversity of target genetic loci was reduced to favor specific traits, and likewise, other loci disappeared (4). Genes associated with desirable phenotypes underwent a diversity loss, where only the alleles of interest were spread in the progenies. It has commonly been hypothesized that with a decrease in genetic diversity, the plant suffers a direct effect on the ability to establish beneficial associations with rhizosphere microbes, and that intensive agricultural practices and the expansion to previously unexplored cultivation regions have weakened certain plant-microbe interactions, which were distinctive and important to wild varieties (5). This widely accepted hypothesis has recently been challenged by larger-scale studies comparing wild and domesticated varieties of multiple plant species. A meta-analysis by Hernández-Terán et al. summarized several studies comparing the microbiome of wild and domesticated plants grown in the same bulk soil and showed that the alpha diversity of the microbial community was higher in the domesticated plants than in their wild relatives, including wheat, tomato, and cotton. Moreover, maize and legumes generally presented higher diversity among their wild counterparts (6). Furthermore, the group acknowledged that host genotype accounted for lesser microbial variation between plant species, and that intra-species variability was relatively high, especially when considering different initial soils. These findings suggest that the effect of plant domestication on rhizosphere microbiomes cannot be generalized as it appears to be species-specific and soil-dependent. Instead of broad shifts in richness, the effect of domestication on the microbiome of a given plant may manifest in more functionally relevant ways that mediate the plant's interaction with the changing environment. Moreover, the microbiome can be affected through artificial selection of plant traits that are involved in plant-microbiome interactions, such as through the direct selection for seed size and nutrient composition, or indirectly through changes in exudate profiles and plant defense capabilities (7).

Rhizosphere microbiome assembly is widely recognized as a filtering mechanism through which the plant selects its associated microbes, offering an alternative explanation for the microbiome differences observed among plant varieties. If plants have undergone a progressive loss of microbiome-selecting genes, this would result in impaired microbiome-filtering abilities, which may explain the higher microbiome diversity observed in several modern plants (7). It is now established that the microbiomes of wild varieties and their domesticated counterparts are significantly different (8). Yet, the effect of genotype on rhizosphere communities of wild varieties and domesticated hosts might remain subtle (9).

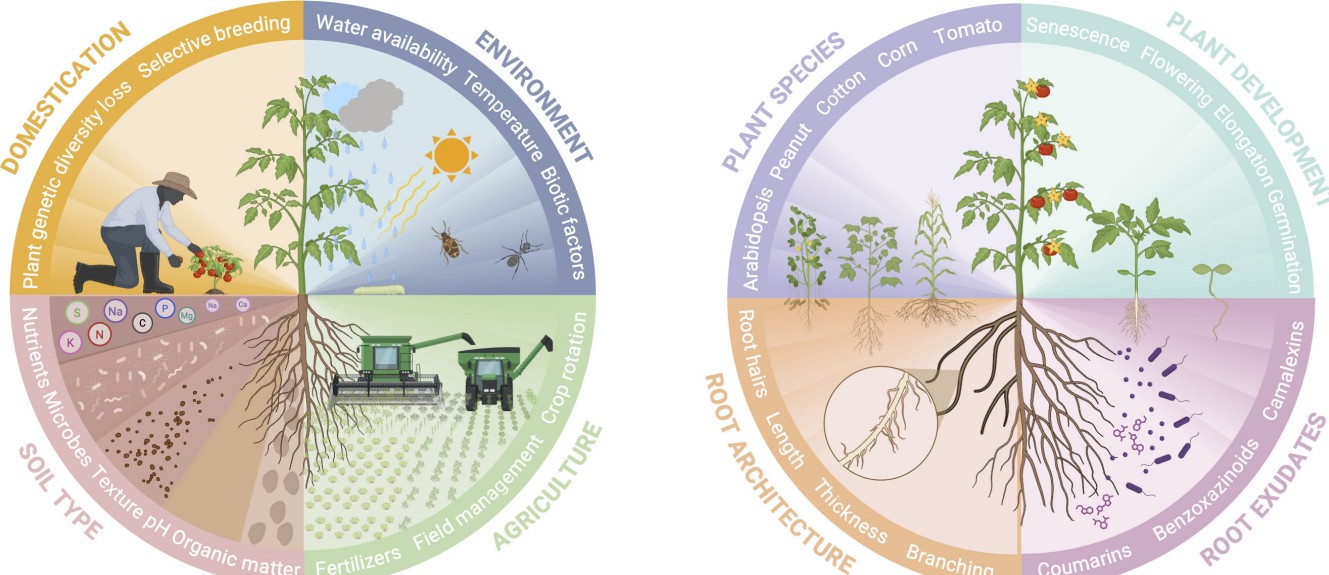

**FIG 1** External and internal plant-specific factors influencing the rhizosphere microbiome composition and diversity. (Left) Environmental factors, including soil type, agricultural context, and evolutionary history involved in shaping the rhizosphere microbiome. (Right) Plant genotype traits and plant developmental stage, including root patterns and exudation profiles controlling the rhizosphere composition and functionality. Created in BioRender.com.

## THE COMBINED ROLE OF ENVIRONMENTAL AND GENOTYPIC FACTORS IN SHAPING THE MICROBIOME

In addition to the plant genotype influencing plants' rhizosphere microbiomes, it is apparent that the environment, and in particular soil type, is often a key determinant of rhizosphere microbiome changes (10, 11). Soil type is determined by a combination of abiotic factors, such as texture, organic matter content, pH, nutrients available for the plant, and biotic factors, particularly the original microbiome present in the soil. Plant roots interact with the surrounding soil to form stronger attachments with a subset of the native soil microbiome; therefore, the resulting rhizosphere microbiome depends on the pre-existing microbial community at a given site.

Along with soil type, climatic conditions and human intervention are likewise involved in shaping the plant microbiome. Water availability and temperature are basic determinants of plant health and growth, and also influence soil communities by creating different environmental niches and habitats. Plant stress caused by such abiotic factors has consistently been shown to affect the rhizosphere microbiome. Moreover, human intervention in the form of various agricultural practices, including fertilization, and application of chemical pesticides is also a driver of plant microbiome changes in agricultural settings.

It is difficult to fully determine whether the environment or the plant genotype has more influence on the soil microbiome surrounding the plant. Each study may report a slightly different contribution of each component depending on the growth setup used. Still, several studies reported comprehensive analyses of the plant microbiome including both soil and genotype factors. Among these, many compared the rhizosphere microbiomes of wild and modern crop varieties grown in different soil types - typically agricultural soils, and those retrieved from the plants' original habitat, known as native soils. For instance, the rhizosphere microbiome of the common bean (*Phaseolus vulgaris*) was found to be more affected by the plant genotype when grown in agricultural soil than in native soil. However, a higher rhizobacterial diversity was found in agricultural soil overall, as well as in the modern bean varieties irrespective of the soil type (12). This illustrates that in a natural setting, genotype and environment are necessarily

intertwined, and that the relative contribution of each factor toward the rhizosphere microbiome is context-dependent (Fig. 1).

Given the significant influence of soil type and other environmental factors on the composition of the plant rhizosphere microbiome, isolating the genotypic component can be challenging. Plant genotype, an inherently complex factor, is thought to "tune" the microbiome, adding a substantial layer of intricacy to this research (13). Therefore, the direct influence of plant genotypes on their microbiome requires careful assessment and consideration of a multitude of aspects that are illustrated in this review.

## CORE RHIZOSPHERE MICROBIOME

The inextricable relationship between plants and their microbiome is well exemplified by the concept of a core microbiome. While the exact definition of a core microbiome might vary between studies, it generally describes the subset of plant-associated microorganisms that are consistently found in all plants of a certain species, within a certain population, or more broadly across populations (14). Given this consistent association, it is generally assumed that these microorganisms have significant functional importance on the plant and are a cornerstone of the plant's interactions with its microbiome (14). Moreover, the presence of a unique core microbiome shared within a plant taxonomic group implies that the dynamics shaping the plant's most closely connected microbial community are to some extent influenced by the plant's genotype. While there is growing interest in describing plant microbiomes, and particularly their core components, it is still challenging to elucidate the mechanisms by which plants interact with these microorganisms and what determines a core microbial composition, in contrast with a conditionally assembled microbiome. Identifying the core microbiome of a plant species can serve as a good starting point for exploring the correlation of plant-associated microbiomes with plant genetics and phylogeny. As an example, two common bean varieties, Eclipse and CELRK, share a common core microbiome including 258 bacterial and archaeal taxa and 13 fungal taxa, even in different growing regions. Proteobacteria, and in particular *Arthrobacter*, are among the most abundant and persistent components of the shared bacterial microbiome, present across all growing locations. Furthermore, taxa unique to either variety were limited and inconsistently present. In this case, the difference between the rhizosphere microbiomes could not be attributed to plant genotype, rather to the plant cultivation condition (15).

Core microbiomes tend to be more distinct between more phylogenetically distant plant species; however, it is unclear whether the microbiome composition is mainly shaped by the environment or the plants' own genetic features. Yeoh et al. examined the root microbiome of 31 plants naturally growing across six consecutive vegetation types. While they identified a significant effect of soil type on the rhizosphere microbiome, plant phylogeny was also found to correlate with root community composition, which was not observed in the bulk soil samples, suggesting that root microbiomes at these sample sites were uniquely shaped through evolution with their specific host. In addition to host-specific microbiome features, the authors further highlighted the presence of a shared subset of microbes between all plant phyla, a phylogeny-wide core microbiome indicating that some tight plant-bacteria associations may have been established long before the evolution of more recent plant lineages (16).

Nevertheless, it remains elusive whether the core microbiome compositions truly mirror the genetic signature of each plant species, or instead, the dynamically distributed members in fact embody each plant species' unique ability to interact with its environment.

## THE ROLE OF ROOT EXUDATES IN SHAPING THE RHIZOSPHERE MICROBIOME

Root exudates are low-molecular-weight compounds secreted by plants, which act as vital substrates for surrounding soil microorganisms, providing them with nutrients and carbon sources (17). Root exudates comprise a diverse range of compounds, including sugars, amino acids, organic acids, phenols, and secondary metabolites (8). The plant

exudation profile is highly dynamic, varying across species, developmental stages, soil types, and environmental factors (18). These root exudates have versatile roles as signaling molecules. They act as attractants and stimulators of beneficial microbes, while also serving as inhibitors and repellents for non-beneficial microbes. Therefore, root exudates gain increasing attention due to the growing evidence of their role in determining the assembly and diversity of the root-associated microbial communities (19). For instance, benzoxazinoids (BXs), the indole-derived secondary metabolites produced by the roots of the *Poaceae* grass species like maize, wheat, and rye (20), have attracted interest in the scientific field due to their dual role acting as biocidal against some bacteria and fungi, while recruiting plant growth-promoting bacteria (19, 20). Cotton et al. demonstrated that mutations in BX biosynthesis genes can alter the maize root rhizobiome, highlighting the role of these genes as part of the plant signaling machinery that shapes the root microbiome (20).

Similarly, Kudjordjie and colleagues correlated BX levels in maize roots with the plant genotype, demonstrating that BXs shape microbial clusters, influencing root fungi more significantly than bacteria. BXs were found to negatively correlate with specific plant pathogens, like *Fusarium* species, known to cause maize diseases. Additionally, they highlighted the role of BX in promoting the activity of beneficial microbes like *Pseudomonas* and *Burkholderia* species, which contribute to plant growth and defense (19).

Recent studies have emphasized the critical role of plant-exudate myo-inositol in shaping the bacterial rhizosphere community. Research by O'Banion et al. showed how myo-inositol transport influenced the composition and behavior of bacterial communities. This metabolite was found to serve as a modulator of bacterial colonization phenotypes, like swimming motility (21). Furthermore, Sánchez-Gil et al. investigated the conserved *iol* gene cluster in the *Pseudomonas* genus, which enables these bacteria to metabolize myo-inositol effectively. The presence of this gene cluster enhances, for example, *Pseudomonas*' rhizosphere competitiveness, facilitating colonization and interaction with plant hosts (22). Taking this altogether, the plant-exuded myo-inositol is an example of the selective attraction of specific microbes by the plant, where plants create niches that promote the establishment of beneficial microbial populations (23).

In addition to their regulatory roles, root exudates serve as a defense mechanism (18). For example, many plant species secrete coumarins, which act as antimicrobial compounds and as iron-mobilizers, promoting the growth and abundance of microbes that increase the availability of iron for the plant (24). Another example is camalexins, antimicrobial compounds produced by plants, including *Arabidopsis thaliana* (Arabidopsis), in response to pathogens or beneficial microbes. Camalexins can selectively inhibit pathogens like *Burkholderia glumae* PG1 in leaves, while allowing the growth of *Pseudomonas* sp. CH267 (25). These examples illustrate the ability of plants to mobilize and vary the exudate concentration to modulate their microbiomes.

The plant immune system consists of an intricate network of pathways that restricts microbial colonization, producing antimicrobial compounds and ensuring the surveillance of specific microbes. Root exudates serve as dynamic intermediaries regulated in part by phytohormones such as salicylic acid (SA), jasmonic acid (JA), and ethylene. These hormones modulate systemic defense responses and the composition of root exudates, which in return, shape the microbial recruitment in the rhizosphere. Lebeis et al. demonstrated that mutations in the biosynthesis pathways of these phytohormones, particularly SA, led to significant differences in the microbiome root composition of Arabidopsis, demonstrating that SA is involved in the exclusion of certain bacterial taxa (26). Similarly, Carvalhais et al. described that alterations in the JA signaling directly affects the root exudate composition and, consequently, the bacterial and archaeal communities in the rhizosphere (27).

Understanding the effects of root exudates on the rhizosphere microbiome requires considering how crop domestication shaped plant evolution, including shift in genetic and phenotypic diversity between wild relatives and modern varieties. For example,

successive wheat genotypes across domestication stages exhibit differential sugar exudation patterns and phenotypic divergence (28). Wild cultivars show a higher stress resistance, whereas modern varieties produce more aboveground biomass and significantly higher sugar levels. This suggests that modern cultivars may have a reduced capacity to modulate sugar exudation compared with their wild relatives (28). These differences across domesticated and undomesticated varieties become more significant when linking these exudation patterns to shifts in the rhizosphere microbiome. A meta-analysis investigating the relationship between domestication and root microbiome composition compared *Cardamine hirsuta* as a wild species and Arabidopsis as the modern variant. This comparison revealed that wild relatives of crop plants generally secrete fewer simple sugars and host a higher abundance of beneficial Bacteroidetes. In contrast, domesticated cultivars tend to harbor more Actinobacteria and Proteobacteria (13). These findings suggest that domestication has altered the root exudation patterns with consequences for root microbiome composition (13).

Evolutionary experiments examining the genetic diversity and exudation differences across ten wheat genotypes, representing critical steps in the domestication of tetraploid wheat (29), demonstrated that different chemical metabolites affect the microbial community dynamics. Among the root exudate compounds, sucrose and organic acids were proposed to directly or indirectly regulate the microbial growth in the root (29). Importantly, human-directed selection of specific varieties drove the domestication of root exudate profiles, with certain metabolites being inherited through the selected genotypes.

In addition to domestication, the plant developmental stage has been hypothesized to influence root exudation. The exudate composition of oat plants (*Avena*) varies significantly over the plant's life cycle, with early-stage plants producing a higher concentration of sucrose and homoserine, which are important for symbiotic interactions and defense (30). As the plant matures, the variety of exudates increases, prioritizing defense strategies above growth. During senescence, plant hormone production including that of indole-3-acetic acid and abscisic acid increases. The researchers proposed that the observed shifts in rhizosphere microbiome patterns could result from the plant's genetic capability to modify the composition of root exudates over time in response to environmental and developmental cues (30).

The influence of soil type on root exudation and microbiome composition is another critical environmental factor that shapes plant-microbe interactions (8). A study on lettuce (*Lactuca sativa L. cv. Tizian*) grown in different soil types revealed that variations in exudate compositions were predominantly quantitative rather than qualitative for low-weight sugars. In this study, despite the absence of external stress factors like pH extremes or nutrient deficiencies, soil-specific characteristics still significantly influenced exudation patterns. These changes were followed by shifts in the composition of the rhizosphere microbiome, showcasing a direct link between soil physicochemical properties, root exudation, and microbial community structure (31).

While the research on the effects of domestication, soil type, and developmental stage on root exudation is promising, much remains to be uncovered. The relationships between plant genotype, root exudate composition, and microbiome assembly are highly complex and often indirect. For example, a single mutation might not directly alter root exudate composition or the microbiome; instead, these changes might involve environmental factors and a series of cascade effects through multiple metabolic steps and signaling pathways. Therefore, there is a current need for comprehensive studies that consider the entire network of interactions between plant genetics, environmental factors, and the soil microbiome.

## THE ROLE OF ROOT ARCHITECTURE IN SHAPING THE RHIZOSPHERE MICROBIOME

When studying the plant rhizosphere microbiome, a key debate revolves around how the microbiome differs not only between distinct plant species but also between different

genotypes of the same plant species. This discussion emphasized a need to "go back to the roots," a renewed interest in understanding how evolution and agricultural practices have shaped the differences between old and modern cultivars, particularly how these differences shape the plants' microbiome (17).

Root architecture, which refers to the spatial and temporal configuration of the plant's root system within the soil matrix (17), plays a crucial role in shaping the root microbiome. The root structure complexity encompasses various traits such as root length, density, biomass, branching patterns, and the growth rate of both lateral and axial roots (32). These root architectural traits are crucial for the plant as they directly impact the nutrient and water uptake in soil niches, thereby shaping the diversity and assembly of the root microbiome (32).

A recent study demonstrated that genetic regulators of root hair development influenced the composition of the root-associated microbiome, particularly under drought conditions. Using Arabidopsis genetic mutants that varied in root hair density, the researchers investigated how these changes affected the soil microbiome composition during drought stress. They found that mutants with enhanced drought tolerance exhibited increased production of secondary metabolites and other enriched metabolic pathways, linking root hair development to different metabolic responses during stress. Furthermore, the findings supported the concept of an intrinsic "cry for help" mechanism in plants, where drought stress triggered the assembly of members of the *Rhizobiaceae* family to support plant resilience. These results highlight the importance of root architecture and genetic regulation in the root microbiome composition under environmental stress (33).

Additionally, a study performed by Robertson-Albertyn et al. compared the root hair's bacterial communities from two barley varieties in two different soil types and two backcrossed inbred lines that completely lacked root hairs or showed an interrupted early root hair development. They demonstrated how a difference in soil type, particularly in terms of nutrient composition, impacted the microbial community. Moreover, after increasing the taxonomic resolution, the root hair genotype was determinant for the abundant members of the root microbiome. In fact, by analyzing the alpha-diversity of the wild-type and mutants in both soils, they showed how, independent of the soil type, the root hair mutants harbored a lower complexity (34).

Root architecture is not static; it responds to numerous environmental factors like soil structure, oxygen and nutrient availability, water content, temperature, plant genetic traits, and soil microorganisms (9). A promising future direction for agricultural practices could focus on the genes that define the root architecture and morphology. As a result, these breeding efforts could indirectly shape the root microbiomes and potentially enhance plant health and productivity (9). For example, different lettuce root architectures, shaped by selective breeding and domestication, create diverse soil microhabitats that support varied microbial communities. The roots of wild cultivars are capable of penetrating deeper soils compared with modern cultivars, which could be linked to differences in the microbial community composition (31). Another recent study by Rouch et al. examined the difference in root architecture by comparing four modern and four ancient wheat varieties, showcasing a difference in root length, with the ancient varieties growing longer roots (35). Also, wheat cultivars with different root architectures have been proposed to influence the taxonomic and functional microdiversity of *Pseudomonas* (36).

Similarly, Szoboszlay et al. investigated the impact of breeding on the root system architecture and rhizosphere microbial communities in maize, comparing domesticated corn varieties to their wild progenitor, the Balsas teosinte (*Zea mays parviglumis*). Their study revealed significant differences in the bacterial and fungal community compositions of rhizosphere samples. Notably, Balsas teosinte exhibited greater bacterial diversity in its rhizosphere compared to sweet corn. Conversely, sweet corn showed a reduction in fungal diversity, a pattern not observed in teosinte. Since microbial communities have been proven to influence plant health through different mechanisms,

like nutrient cycling and pathogen suppression, this finding underscores the potential benefits of considering microbial community composition and disease resilience during crop breeding (2, 9).

A study performed by Pérez-Jaramillo et al. studied how the relationship between plant genetics and the root architecture patterns of wild and modern accessions of common bean (*Phaseolus vulgaris*) affected the rhizosphere microbiome composition. Based on their results, 13.5% of the total variability of the host-dependent rhizosphere bacterial community composition was defined by the plant genotype. In the case of the wild relatives with thinner roots, there was an increased abundance of Bacteroidetes, in contrast to the higher abundance of Actinobacteria and Proteobacteria present in the thicker roots of modern varieties. Taken together, the researchers showed the complex network relating the plant's history, genotype, and specific root patterns to the bacterial rhizosphere community (12).

Other traits, such as root diameter, depth, and branching, have also been hypothesized to play significant roles (32). In terms of root diameter, thinner roots have been associated with more diverse rhizosphere communities (17). This phenomenon is hypothesized to occur due to a limited root physical surface, leading to improved root exudation and nutrient cycling by the plant, and indirectly, a higher competition among microbes (17). Another architectural component is the root depth, as deeper roots enable plants to reach nutrient-rich niches and encounter a wider range of bacteria taxa present in lower soil layers (32, 37). Enhanced branching increases root surface area, enabling the development of thinner roots that can exploit micro-niches of nutrients (38). Additionally, modifications of root cortical aerenchyma, which is crucial for nutrient and gas exchange (39) and suberin deposits within the root endodermal barrier, affect the flow of iron and salts (40). Overall, these examples showcase dynamic root architectures and their impact on nutrient and water acquisition, which in turn influences the root microbiome.

Understanding the interplay between root architecture and microbial diversity will promote targeted genetic modifications to improve both plant performance and microbial health.

## STUDYING PLANT GENOTYPE-RHIZOSPHERE MICROBIOME ASSOCIATIONS

Identifying plant genetic factors that influence the plant rhizosphere microbiome remains a stimulating open challenge. The expanding knowledge on plant-microbe interactions facilitates the discovery of specific plant genes that affect the microbiome assembly and functioning in the rhizosphere, as well as pathways known to influence the interactions of the plant with its (biotic) environment, such as through root exudates. Utilizing this valuable information, common experimental approaches employ mutated plant lines lacking selected genes of interest to uncover the effect of specific plant genotypic traits on the microbiome. Voges et al. employed a synthetic microbial community (SynCom) composed of Arabidopsis commensals and analyzed the root microbiome composition formed on different Arabidopsis mutants with impaired biosynthesis of a variety of secreted metabolites. Lack of coumarin production in the Arabidopsis mutant line *f6'h1-1* resulted in increased abundance of a *Pseudomonas* sp. strain compared to wild-type Arabidopsis under iron-limiting conditions, probably associated with recognized antimicrobial activity of coumarins produced by iron-deficient plants (41).

The use of synthetic microbial communities (SynComs) represents a reductionist approach to observe the ecological implications determined by different plant hosts (42). SynComs are artificially assembled consortia with reduced complexity compared with natural microbiomes, facilitating the study of ecological dynamics, including microbe-microbe interactions, as well as host-microbe interactions (43, 44). For instance, a SynCom may be created to represent a simplified core microbiome of a plant host. Wippel et al. tested the root microbiome assembly of two communities representing the microbiome of *Lotus japonicus* and Arabidopsis and observed a heightened ability to

colonize each host by commensal bacteria originally derived from that host (45). Thanks to their reduced complexity, SynComs can be easily inspected to detect abundance changes induced by host-specific features. It is not yet understood whether a plant's genetic signature broadly influences the entire rhizosphere microbiome or primarily affects a small subset of microbial species. If the latter is true, there is a risk of underestimating the impact of plant genetics when analyzing the whole microbiome (46). Focusing on a simplified microbial community, instead, could offer enhanced resolution to uncover these effects.

Most approaches used to link plant genetic features to their microbiome are limited to the existing knowledge of complex plant traits, which might be difficult to relate to the microbiome in a predictable way. Genome-wide association studies (GWAS) are emerging as an innovative way of interpreting genetic data in an ecological context (47). GWAS take genomic data and relevant phenotypic information as input to find genes associated with a trait of interest. In one of its most fascinating forms, microbiome data can be used as input to GWAS to identify correlations between plant genetic loci and the abundance of microbial groups in the plant's microbiome. Through the application of GWAS using 16S amplicon data of rhizosphere samples, several bacterial taxa were found to be highly heritable in sorghum, meaning that their abundance is highly host genotype-dependent. Further, Deng et al. identified a genetic locus likely involved in root-localized activity, which correlated with the abundance of specific taxa (48). Yet, the identification of specific loci shaping the rhizosphere microbiome can be hindered by the intrinsic complexity governing their expression, as multiple genetic components and environmental factors may be involved in their activity. The intricacies of how plant genetic traits regulate rhizosphere microbiome assembly are the product of their extended evolution over millions of years. Wagner et al. compared the rhizosphere microbiomes of inbred and corresponding F1 hybrid lines of maize, hypothesizing that heterosis—the genetic phenomenon in which traits of hybrid lines exceed both parent lines—may similarly manifest in microbiome features. Traces of microbiome heterosis were found in individual hybrid lines, with a stronger host-genotype effect detected on rhizosphere microbiomes (46). Studying the patterns of inheritance of microbiome-related traits can help effectively integrate them into current breeding strategies.

## CONCLUSIONS

The importance of a plant microbiome toward plant health and fitness is already well established, leading to growing interest in understanding the dynamics shaping the plant microbiome for crop growth promotion. Plant genetics is an important factor which can influence the rhizosphere microbiome, most notably through regulating root exudation and root architecture. However, the complexity of microbiome-relevant plant traits is far from resolved. Disentangling the roles of plant genetics and environmental factors in shaping the rhizosphere microbiome remains a stimulating challenge, as does the discovery and investigation of specific key traits, their expression, and heritability. Through plant domestication and crop breeding, plant traits that were originally shaped by natural selection undergo alterations. Consequently, any genetic determinants of microbiome variation may similarly be inherited or lost through the selection of desired plant traits. Incorporating knowledge of how plant genetics affect the rhizosphere microbiome can enhance the development of sustainable agricultural solutions harnessing plant microbiomes.

## ACKNOWLEDGMENTS

We thank Pingtao Ding for his thoughtful review and suggestions.

The lab of Á.T.K. is supported by a start-up fund from the Institute of Biology Leiden, the Novo Nordisk Foundation within the INTERACT project of the Collaborative Crop Resiliency Program (NNF19SA0059360), and the European Union (ERC, MicroClock, 101166968). Views and opinions expressed are, however, those of the author(s) only and

do not necessarily reflect those of the European Union or the European Research Council Executive Agency. Neither the European Union nor the granting authority can be held responsible for them.

## AUTHOR AFFILIATIONS

[1]Institute of Biology, Leiden University, Leiden, the Netherlands
[2]DTU Bioengineering, Technical University of Denmark, Kongens Lyngby, Denmark

## AUTHOR ORCIDs

María Negre Rodríguez  http://orcid.org/0009-0009-7827-1234
Adele Pioppi  http://orcid.org/0009-0005-7400-8464
Ákos T. Kovács  http://orcid.org/0000-0002-4465-1636

## FUNDING

| Funder | Grant(s) | Author(s) |
| --- | --- | --- |
| Novo Nordisk Fonden | NNF19SA0059360 | Ákos T. Kovács |
| European Research Council | 101166968 | Ákos T. Kovács |

## AUTHOR CONTRIBUTIONS

María Negre Rodríguez, Conceptualization, Visualization, Writing – original draft | Adele Pioppi, Conceptualization, Visualization, Writing – original draft | Ákos T. Kovács, Conceptualization, Funding acquisition, Supervision, Writing – review and editing

## ADDITIONAL FILES

The following material is available online.

Open Peer Review

**PEER REVIEW HISTORY (review-history.pdf).** An accounting of the reviewer comments and feedback.

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
