## [Reviewer comments · mSystems]

The role of plant host genetics in shaping the composition and functionality of rhizosphere microbiomes

María Negre Rodríguez, Adele Pioppi, and Ákos T. Kovács

Corresponding Author(s): Ákos T. Kovács, Universiteit Leiden

Review Timeline:

Submission Date:	December 24, 2024
Editorial Decision:	March 18, 2025
Revision Received:	May 5, 2025
Accepted:	June 6, 2025

Editor: Van Schepler-Luu

Reviewer(s): Disclosure of reviewer identity is with reference to reviewer comments included in decision letter(s). The following individuals involved in review of your submission have agreed to reveal their identity: Xiaoping Li (Reviewer #1)

Transaction Report:

DOI: <https://doi.org/10.1128/msystems.00041-24>

Re: mSystems00041-24 (The role of plant host genetics in shaping microbiome composition and functionality)

Dear Prof. Ákos T. Kovács:

The manuscript effectively reviews the influence of plant genetics on the rhizosphere microbiome. The reviewers recommend revising the manuscript to better highlight the focus on the rhizosphere microbiome rather than the entire plant microbiome. Reviewer 1 suggests adjusting the title and reorganizing sections, particularly by revising the "Plant genotype and environment jointly influence the microbiome" section to emphasize plant genotype and presenting environmental factors as subsections. Both reviewers highlight the need for clearer connections and refined language throughout, especially in terms of explaining the relationships between plant genotype, microbiome diversity, and environmental influences. Additionally, Figure 1 should be expanded to include more nuanced factors, such as root exudates and plant immunity, with supporting studies to avoid oversimplification.

Revision Guidelines

- Upload point-by-point responses to the issues raised by the reviewers in a file named "Response to Reviewers," NOT in your cover letter.
- Upload a compare copy of the manuscript (without figures) as a "Marked-Up Manuscript" file.
- Upload a clean .DOC/.DOCX version of the revised manuscript and remove the previous version.
- Each figure must be uploaded as a separate, editable, high-resolution file (TIFF or EPS preferred), and any multipanel figures must be assembled into one file.

Minireviews are not subject to publication charges.

Author Bios: We encourage you to submit a biographical sketch of each author (limit of 150 words) along with a photo to be published at the end of your article. You can submit these with your modified manuscript.

Figures Enhancement: ASM has engaged a professional science illustrator, Patrick Lane of ScEYence Studios, to work with minireview authors at the modification stage to generate improved figures that are uniform throughout the journal. This art enhancement service is free of charge to authors of minireviews and full-length reviews, and turnaround time is fast. I think you will be pleased with the results. Please contact Patrick on receiving this letter. Complete contact information for Patrick and further instructions are posted at <https://journals.asm.org/pb-assets/pdf-text-excel-files/graphical-enhancement-support.pdf>.

Sincerely,
Van Schepler-Luu
Editor
mSystems

Reviewer #1 (Comments for the Author):

It is a well-written manuscript reviewing the current knowledge on the plant genetics/genotypes shaping the microbiome in the below-ground compartments.

As the focus is on the rhizosphere and root microbiome, I think the title may need to be revised to better reflect that.

I would suggest that subtitles be added under each section to better organize the manuscript coherently.

I would also think it is a better idea to revise the section of "Plant genotype and environment jointly influence the microbiome" to "Influence of the plant genotype on the microbiome" something along the line; therefore, revising the environment factors as subsections. You may need to REORGANIZE and CONSOLIDATE those parts to bring the spotlight to 'plant genotype' better while demonstrating that domestication, agricultural management, and environmental conditions are also part of the process, thus echoing your title.

Each section needs work to tighten the main messages. For example, while the message of L76-80 shows that soil type is a determinant factor of the rhizosphere microbiome, the remaining paragraph discusses the formation of rhizosphere microbiome depending on the previous microbial composition.

Other comments:

L80, "importantly" and "depends on" sound redundant

L124, A) Your focus here is the core microbiome, but the example describes the 'rhizosphere microbiome'. B) Unclear about 'a minor correlation with plant phylogeny was also found' - which correlated with which. C) Unclear about 'plants co-evolved with their microbiome at the same sites' based on your evidence here.

L305, 'pathogen suppression capabilities' needs to be introduced properly, otherwise it confused me as none of the paragraphs above connect me to this point.

Reviewer #2 (Comments for the Author):

This is an overall clearly written and well-structured minireview of the influence of plant genetics on rhizosphere microbiome. However, the language should be modified to clarify that the review is really about rhizosphere microbiomes, not the entire plant microbiome.

Figure 1 indicates that modern cultivars have reduced microbiome diversity and reduced root biomass compared to wild relatives. This figure and takeaway seems overly simplistic and not universal across different plants and studies discussed in the review. It would be interesting to bring in the differences in root exudates, plant immunity, etc, here, and potentially to link relevant studies supporting what is highlighted in the figure.

Minor comments:

Line 86-89. The way these 2 sentences are written is difficult to understand. "more affected by plant genotype than by native soil" - it is not clear how this distinction is made. It would help if a little more information was provided about the specific soils compared.

In the core microbiome section and most other sections, all examples appear to be specific to the rhizosphere microbiome. Throughout the minireview there is discussion of the plant microbiome, but the authors are almost always talking exclusively about the rhizosphere.

Line 174 - *Arabidopsis* should be italicized.

Lines 194-198. The flow of these sentences is not clear. There should be more connection and clarity here.

Figure 2 typo "organic mater" should be "organic matter"

Editorial comments:

The manuscript effectively reviews the influence of plant genetics on the rhizosphere microbiome. The reviewers recommend revising the manuscript to better highlight the focus on the rhizosphere microbiome rather than the entire plant microbiome. Reviewer 1 suggests adjusting the title and reorganizing sections, particularly by revising the "Plant genotype and environment jointly influence the microbiome" section to emphasize plant genotype and presenting environmental factors as subsections. Both reviewers highlight the need for clearer connections and refined language throughout, especially in terms of explaining the relationships between plant genotype, microbiome diversity, and environmental influences. Additionally, Figure 1 should be expanded to include more nuanced factors, such as root exudates and plant immunity, with supporting studies to avoid oversimplification

Response: We thank you and the two reviewers for the helpful comments and guidance. We have adapted the title and commented sections according to these recommendations. We have also improved the figures and merged them as panel A and B, to make those more uniform and include more nuanced factors, such as root exudates and plant immunity.

Reviewer #1 (Comments for the Author):

It is a well-written manuscript reviewing the current knowledge on the plant genetics/genotypes shaping the microbiome in the below-ground compartments.

As the focus is on the rhizosphere and root microbiome, I think the title may need to be revised to better reflect that.

I would suggest that subtitles be added under each section to better organize the manuscript coherently.

Response: Thank you for your suggestion. We have now adapted the title to emphasize the rhizosphere microbiome as the main topic. We have also adapted the current section titles to clarify the scope of each section.

Revised part:

- New title: "The role of plant host genetics in shaping the composition and functionality of rhizosphere microbiomes"
- Section titles: "The rhizosphere microbiome"; "Domestication: How the plant genotype influences the rhizosphere microbiome"; "The combined role of environmental and genotypic factors in shaping the microbiome"; and "Core rhizosphere microbiome".

I would also think it is a better idea to revise the section of "Plant genotype and environment jointly influence the microbiome" to "Influence of the plant genotype on the microbiome" something along the line; therefore, revising the environment factors as subsections. You may need to REORGANIZE and CONSOLIDATE those parts to bring the spotlight to 'plant genotype' better while demonstrating that domestication, agricultural management, and environmental conditions are also part of the process, thus echoing your title.

Response: Thank you for your suggestions. We have reorganized the content into a first section focusing on domestication as an example of how the plant genotype influences the rhizosphere microbiome, followed by a section on environmental factors which are involved in microbiome shifts. Here, we intend to acknowledge that the combination of genotypic and

environmental factors is what ultimately shapes the microbiome. As for the overall structure of the review, we aim to first contextualize the genotype factor in the form of the plant's evolutionary history and its surrounding environment, to then go deeper into the main plant inner factors that are directly shaping the rhizosphere microbiome.

Revised part: L. 54 to 141 (note, lines refer to the version with track changes)

Each section needs work to tighten the main messages. For example, while the message of L76-80 shows that soil type is a determinant factor of the rhizosphere microbiome, the remaining paragraph discusses the formation of rhizosphere microbiome depending on the previous microbial composition.

Response: We understand why this appears confusing. We have clarified that the native soil microbiome is an important biotic factor which characterizes each soil type, along with abiotic factors which we have now listed as well.

Revised part: Soil type is determined by a combination of abiotic factors, such as texture, organic matter content, pH, and nutrients available for the plant; and biotic factors, particularly the original microbiome present in the soil. Plant roots interact with the surrounding soil to form stronger attachments with a subset of the native soil microbiome; therefore, the resulting rhizosphere microbiome depends on the pre-existing microbial community at a given site.

Other comments:

L80, "importantly" and "depends on" sound redundant

Response: Thank you for pointing this out. We have addressed this in the previous comment. L124, A) Your focus here is the core microbiome, but the example describes the 'rhizosphere microbiome'.

Response (A): We recognize that this appears confusing. The example is intended to show that, in addition to plant specific core root microbiomes, a core microbiome of shared rhizosphere microbes can also be identified across plant species, widening the notion of "core microbiome" to a higher taxonomic level of abstraction.

B) Unclear about 'a minor correlation with plant phylogeny was also found' - which correlated with which.

Response (B): We have now clarified that Yeoh *et al.* found a correlation between plant phylogeny and root microbiome composition, albeit minor compared to the effect of soil type on root microbiome composition.

C) Unclear about 'plants co-evolved with their microbiome at the same sites' based on your evidence here.

Response (C): The authors of the cited paper report "a small but significant correlation between ordinations summarizing variation in root bacterial community composition and plant phylogenetic distance" and further express that this indicates unique coevolution of root communities with their hosts.

Revised part (addressing A, B, and C): While they identified a significant effect of soil type on the rhizosphere microbiome, plant phylogeny was also found to correlate with root community composition, which was not observed in the bulk soil samples, suggesting that root microbiomes at these sample sites were uniquely shaped through evolution with their

specific host. In addition to host-specific microbiome features, the authors further highlight the presence of a shared subset of microbes between all plant phyla, a phylogeny-wide core microbiome indicating that a portion of tight plant-bacteria associations may have been established long before the evolution of more recent plant lineages.

L305, 'pathogen suppression capabilities' needs to be introduced properly, otherwise it confused me as none of the paragraphs above connect me to this point.

Response: Thank you for pointing this out. To clarify the relevance of pathogen suppression and give a connection to the plant-microbiome context, we have revised the paragraph by adding a brief introductory sentence and a citation where the authors discussed the role of microbial communities in disease resistance and a brief introduction.

Revised part: Conversely, sweet corn showed a reduction in fungal diversity, a pattern not observed in teosinte. Since microbial communities have been proven to influence plant health through different mechanisms, like nutrient cycling and pathogen suppression, this finding underscores the potential benefits of considering microbial community composition and disease resilience during crop breeding (6).

Reviewer #2 (Comments for the Author):

This is an overall clearly written and well-structured minireview of the influence of plant genetics on rhizosphere microbiome. However, the language should be modified to clarify that the review is really about rhizosphere microbiomes, not the entire plant microbiome.

Figure 1 indicates that modern cultivars have reduced microbiome diversity and reduced root biomass compared to wild relatives. This figure and takeaway seems overly simplistic and not universal across different plants and studies discussed in the review. It would be interesting to bring in the differences in root exudates, plant immunity, etc, here, and potentially to link relevant studies supporting what is highlighted in the figure.

Response: Thank you for your input. We recreated the illustrations and merged into a single figure to include all these above suggestions. The current figure with the two panels represents a more uniform style. Now, the first panel focuses on external factors influencing the microbiome, while the second panel focuses on internal, plant-specific factors involved in shaping the microbiome. Domestication has now been addressed in a more nuanced way in its own section and has been included as an externally driven process in panel A, while root exudates are mentioned in panel B.

Minor comments:

Line 86-89. The way these 2 sentences are written is difficult to understand. "more affected by plant genotype than by native soil" - it is not clear how this distinction is made. It would help if a little more information was provided about the specific soils compared.

Response: Thank you for the advice on how to clarify this sentence. We have rephrased and added a short introductory sentence on soil types and how they are employed in such studies.

Revised part: Still, several studies report comprehensive analyses of the plant microbiome including both soil and genotype factors. Among these, many compare the rhizosphere microbiome of wild and modern crop varieties when grown in different soil types, typically

agricultural soil, and soil retrieved from the plants' original habitat, termed native soil. For instance, the rhizosphere microbiome of the common bean (*Phaseolus vulgaris*) was found to be more affected by the plant genotype when grown in agricultural soil than in native soil.

In the core microbiome section and most other sections, all examples appear to be specific to the rhizosphere microbiome. Throughout the minireview there is discussion of the plant microbiome, but the authors are almost always talking exclusively about the rhizosphere.

Response: We recognize that this was a recurrent issue throughout the manuscript. We specified for each section that the focus is on the rhizosphere microbiome.

Line 174 - *Arabidopsis* should be italicized.

Response: *Arabidopsis* should be italicized if it refers to a genus name. However, here, we refer to the species *Arabidopsis thaliana*, which is only italicized if the full Latin name is used. To make it obvious we are not referring to the genus (hence not italicized), we have added *Arabidopsis* without being italicized after the first time we mention *Arabidopsis thaliana*. This is a general practice by plant biologist, hence, we were suggested not to italicize *Arabidopsis* if we do not refer to the genus.

Lines 194-198. The flow of these sentences is not clear. There should be more connection and clarity here.

Response: Thank you for the observation. We have modified this section and made a clearer connection between domestication and differences in the phenotype and sugar exudation patterns.

Revised part: To fully comprehend the effects of root exudates in the rhizosphere microbiome, the historical context of crop domestication should also be considered, including the comparison of the trait and genetic diversity of wild relatives with modern crops. For example, various wheat genotypes representing the main stages of wheat evolution differing in their sugar exudation intensity, present variations in their phenotype (25). Wild cultivars show a higher stress resistance, whereas modern varieties produce more aboveground biomass and significantly higher sugar levels. This suggests that modern cultivars may have a reduced capacity to modulate the sugar exudation compared with their wild relatives (25). These differences across domesticated and undomesticated varieties become more significant when linking these exudation patterns to shifts in the rhizosphere microbiome. A meta-analysis investigating the relationship between domestication and root microbiome composition compared *Cardamine hirsute* as a wild species and *Arabidopsis* as the modern variant.

Figure 2 typo "organic mater" should be "organic matter"

Response: Thank you for pointing out this mistake. We have corrected the spelling of the figure.

Re: mSystems00041-24R1 (The role of plant host genetics in shaping the composition and functionality of rhizosphere microbiomes)

Dear Prof. Ákos T. Kovács:

Thank you very much for your patience. Please revise the manuscript to incorporate the suggestion by review 1.

Your manuscript has been accepted, and I am forwarding it to the ASM production staff for publication. Your paper will first be checked to make sure all elements meet the technical requirements. ASM staff will contact you if anything needs to be revised before copyediting and production can begin. Otherwise, you will be notified when your proofs are ready to be viewed.

Sincerely,
Van Schepler-Luu
Editor
mSystems

Reviewer #1 (Comments for the Author):

Thank you for the revision to enhance the readability of the manuscript. I have only a few minor comments and suggestions to improve its clarity in several places.

L72, Many long sentences in this manuscript still read 'clunky', like this one: "A meta-analysis by Hernandez-Terán et al. summarized several studies where wild and domesticated relative plants were inoculated with the same bulk soil and showed that most of the studied domesticated plants' microbiomes had in fact a higher alpha diversity than their wild counterparts,

particularly wheat, tomato, and cotton." Can you re-word them? For example, "A meta-analysis summarized several studies comparing the microbiome between wild and domesticated plants grown in the same bulk soil and showed that the alpha diversity of the microbial community was higher in the domesticated plants than in their wild relatives, including wheat, tomato, and cotton."

L73, instead should be used to indicate alternative. Better to use 'on the other hand' or 'in contrast', something like that to refer to the opposite.

L75, may be 'host genotype accounted for...'

L90 to 92, not clear the meaning, needs rewording.

L93, do you have reference here to support 'Due to ... arising between plant varieties'?

L97, what is 'genotype component'?

L110, 'these similarly affect' confuses me. You mean water availability and temperature have the same effects on the microbiome?

L113, fertilization and application of chemical pesticides are part of agricultural practices

L117, reported - check the tense throughout

L119, compared - same as above

L120, remove 'when'

L120, different soil types - typically agricultural soils and those retrieved from plants' original habitats, known as native soils

L123, in-text citation

L123-124, Interestingly to However

L152, specify what makes them the 'key component' - are they the hubs or something else?

L153 to 155, This sentence is not clear, I had to guess the meaning

L185, just one bacterium here?

L182, BX or Bx? Be consistent

L188, "acting as regulators ... system" confuses me here.

L213-222. The message of this paragraph is not clear. It seems to talk about plant hormones and root exudates rather than network of pathways in the plant immune system.

L226, not clear

L223-232, is this about historical context of crop domestication or genetic evolution?

L235 to 238, too long and hard to understand

L251, plant hormone production including that of the IAA and ABA increases

L283, do you mean root architectural traits impact nutrient and water uptakes? If not, how do they impact the availability of nutrients and water in the soil

L295-303, belong to the previous paragraph

L329, root diameter seems out of the order here

L368, simpler approach as in what context?

L370, need to explain the 'simplified nature'